# Methanolic Extract of *Piper sarmentosum* Attenuates Obesity and Hyperlipidemia in Fructose-Induced Metabolic Syndrome Rats

**DOI:** 10.3390/molecules26133985

**Published:** 2021-06-29

**Authors:** Sivanesan Raja Kumar, Elvy Suhana Mohd Ramli, Nurul Alimah Abdul Nasir, Nafeeza Mohd Ismail, Nur Azlina Mohd Fahami

**Affiliations:** 1Department of Pharmacology, Faculty of Medicine, Universiti Kebangsaan Malaysia, Jalan Yaacob Latif, Cheras, Kuala Lumpur 56000, Malaysia; sivanesanvez@yahoo.com; 2Department of Anatomy, Faculty of Medicine, Universiti Kebangsaan Malaysia, Jalan Yaacob Latif, Cheras, Kuala Lumpur 56000, Malaysia; elvysuhana@ukm.edu.my; 3Centre for Neuroscience Research (NeuRon), Faculty of Medicine, Universiti Teknologi MARA, Sungai Buloh, Selangor 47000, Malaysia; nurulalimah@uitm.edu.my; 4School of Medicine, International Medical University, Bukit Jalil, Kuala Lumpur 57000, Malaysia; nafeeza06@hotmail.com

**Keywords:** *Piper sarmentosum*, metabolic syndrome, obesity, hyperlipidemia, leptin

## Abstract

Obesity and hyperlipidemia are metabolic dysregulations that arise from poor lifestyle and unhealthy dietary intakes. These co-morbidity conditions are risk factors for vascular diseases. *Piper sarmentosum* (PS) is a nutritious plant that has been shown to pose various phytochemicals and pharmacological actions. This study aimed to investigate the effect of PS on obesity and hyperlipidemia in an animal model. Forty male Wistar rats were randomly divided into five experimental groups. The groups were as follows: UG—Untreated group; CTRL—control; FDW—olive oil + 20% fructose; FDW-PS—PS (125 mg/kg) + 20% fructose; FDW-NGN—naringin (100 mg/kg) + 20% fructose. Fructose drinking water was administered daily for 12 weeks ad libitum to induce metabolic abnormality. Treatment was administered at week 8 for four weeks via oral gavage. The rats were sacrificed with anesthesia at the end of the experimental period. Blood, liver, and visceral fat were collected for further analysis. The consumption of 20% fructose water by Wistar rats for eight weeks displayed a tremendous increment in body weight, fat mass, percentage fat, LDL, TG, TC, HMG-CoA reductase, leptin, and reduced the levels of HDL and adiponectin as well as adipocyte hypertrophy. Following the treatment period, FDW-PS and FDW-NGN showed a significant reduction in body weight, fat mass, percentage fat, LDL, TG, TC, HMG-CoA reductase, and leptin with an increment in the levels of HDL and adiponectin compared to the FDW group. FDW-PS and FDW-NGN also showed adipocyte hypotrophy compared to the FDW group. In conclusion, oral administration of 125 mg/kg PS methanolic extract to fructose-induced obese rats led to significant amelioration of obesity and hyperlipidemia through suppressing the adipocytes and inhibiting HMG-CoA reductase. PS has the potential to be used as an alternative or adjunct therapy for obesity and hyperlipidemia.

## 1. Introduction

Obesity is a chronic non-communicable disease that is widely spread across the globe and threatens the health of mankind. This rising epidemic depicts the extensive changes in society and behavioral patterns of communities over the past decades. Economic growth, modernization, urbanization, and globalization of food markets are some of the important attributes that accelerate the progression of obesity [1]. On the other hand, obesity is associated with an increased risk of developing other serious illnesses like hypertension, hyperlipidemia, heart disease, type 2 diabetes, arthritis, and certain forms of cancer [2]. The prevalence of obesity is at an alarming rate around the world. A report by the World Health Organization (WHO) in 2016 estimated that nearly two billion adults of both sexes worldwide were overweight and, of these, more than half a billion were obese [3].

In addition, the WHO also reported that globally, at least 2.8 million deaths are directly related to being obese or overweight [3]. Being obese contributes to abnormalities in lipid metabolism, which leads to the excessive production of triglycerides and low-density lipoprotein (LDL) [4]. Hyperlipidemia is characterized by an increase in the level of serum triglyceride, LDL, total cholesterol (TC), and a decrease in the level of high-density lipoprotein (HDL) [5]. Raised levels of lipids or cholesterol in the blood increase the risks of heart disease and stroke. In 2008, the WHO reported that the prevalence of raised total cholesterol among adults for both sexes was 39% and over 50% of adults from high-income countries had increased total cholesterol level, which was more than double the level than those in low-income countries [6].

Lifestyle modifications that include proper diet and a decent amount of physical activity have been and are the mainstay in the prevention and treatment of metabolic syndromes. However, as individuals tend to be more occupied at work, they are more unlikely to adhere to the lifestyle intervention. As a result, pharmacological medications such as statins, orlistat, and ezetimibe are considered as the second line of intervention in the treatment and prevention of obesity and hyperlipidemia [7]. Nevertheless, the use of these medications remains controversial as they are associated with various side effects [8,9]. Over the past decades, the administration of modern (synthetic) medicine has provided a significant gain in health care and boosted the quality of living. However, due to the increase in population size, the excessive cost of treatments, an insufficient supply of drugs, side effects, and the development of resistance to certain synthetic drugs, the present drug discoveries in intervention emphasize using plant compounds as a source of medication for a wide variety of modern diseases.

In addition, the WHO also recently estimated that almost 80% of the world’s population relies on traditional medicines for their health care needs and around 21,000 plant species have the potential for being used as medicinal plants [10]. The use of plants in health care may offer an unlimited source of chemical diversity to identify new drug modules as they are less toxic, safe, more efficient, and less expensive [11]. *Piper sarmentosum* (PS), also known as “betel leaves”, is a shade-tolerant, low growing perennial medicinal plant that is widely distributed (1200 species) throughout the pantropical and neotropical regions of the world [12]. PS consists of a wide range of phytochemicals such as xanthophylls, tannins, phenolic compounds, calcium, iron, vitamin B 1, 2, C, E, β-carotene, and β-sitosterol [13,14,15]. Moreover, the plant has been extensively used traditionally to treat a variety of diseases ranging from headaches to inflammatory and rheumatic diseases [12,16]. In addition, PS also poses citrus flavonoid naringin, which has been shown to reduce fat deposition, lipid levels, fasting glucose, oxidative stress, and generation of pro-inflammatory cytokines [12,17].

Due to its strong traditional application, the plant has been broadly studied to provide scientific testimonials for its use in the drug development sector. As PS consists of strong phytonutrients like amides, pyrones, and flavonoids, it poses strong pharmacological properties such as antibacterial, antifungal, antiviral, antioxidant, hepatoprotective, anti-inflammatory, anti-atherosclerosis, and hypoglycemic activities [16]. At present, there is relatively limited information available that is related to the protective effect of PS and its use in alleviating obesity, hyperglycemia, hypertension, and hyperlipidemia concurrently. Therefore, in this study, we investigated the effect of PS toward obesity and hyperlipidemia in fructose-induced metabolic syndrome in a rat model, which will be the foundation of knowledge in the development of PS as a supplementation in preventing and reducing complications associated with metabolic syndrome. The hypothesis of the study was *Piper sarmentosum*, which reduces body weight, fat mass and percentage, cholesterols, leptin, HMG-CoA activity, size of adipocytes, and increased levels of HDL as well as adiponectin.

## 2. Results

### 2.1. Physiological Parameters

Table 1 shows the effect on physiological variables following the eight-week consumption of fructose drinking water and four-week treatment period. The group of rats that underwent fructose induction for eight weeks experienced a significant increase in daily food (rat chow) intake compared to the control group (*p* < 0.05). During the 28 days treatment period, it was found that FDW-PS and FDW-NGN groups showed a significant decrease in their daily food intake compared to the FDW group (*p* < 0.05). However, there were no significant differences found between the FDW-PS and FDW-NGN groups in their daily food intake.

The group of rats that received fructose experienced a significant increase in daily fluid intake compared to the CTRL group (*p* < 0.05). Treatment for 28 days showed that group FDW-PS and FDW-NGN rats had a significant decrease in their daily fluid intake compared to the FDW group (*p* < 0.05). No significant differences were found between the FDW-PS and FDW-NGN groups in their daily fluid intake throughout the treatment. 

The total calorie intake for a day was significantly higher in rats that consumed fructose compared to the CTRL group (*p* < 0.05). After 28 days of treatment, the FDW-PS and FDW-NGN groups of rats had a lower calorie intake compared to the FDW group (*p* < 0.05). There was no significant difference found between the FDW-PS and FDW-NGN groups in their daily calorie intake.

### 2.2. Metabolic Parameters

#### 2.2.1. Body Weight

The higher total calorie intake by rats supplemented with fructose water for eight weeks led to significantly higher weight gain in FDW rats (440.70 ± 3.77 g) compared to the control group (327.90 ± 5.99 g) (*p* < 0.05). After the 28-day treatment period, it was found that the FDW-PS and FDW-NGN rats showed a significant reduction in weight gain at (401.54 ± 4.20 g) and (411.90 ± 1.56 g) compared to the FDW group (*p* < 0.05). Furthermore, the FDW-PS group showed a much more significant reduction in body weight gain compared to the FDW-NGN group (*p* < 0.05) (Figure 1).

#### 2.2.2. Fat Mass and Percentage

Significantly higher body fat mass and percentage were observed in FDW rats (66.21 ± 1.81 g and 14.36 ± 0.20%, respectively) that received fructose for eight weeks compared to the control group (18.38 ± 0.53 g and 8.11 ± 0.52%, respectively) (*p* < 0.05). After the treatment period of 28 days, it was found that FDW-PS and FDW-NGN rats showed a significant reduction in fat mass at (56.53 ± 1.06 g) and (61.02 ± 1.81 g) compared to the FDW group (*p* < 0.05). In addition, the 28 day treatment showed that the FDW-PS and FDW-NGN rats had a significant reduction in fat percentage (11.01 ± 0.31%) and (12.41 ± 0.29%) compared to the FDW group (*p* < 0.05). FDW-PS rats had a more significant reduction in fat mass and percentage compared to the FDW-NGN rats (*p* < 0.05) (Figure 2).

#### 2.2.3. Adipocytokines

FDW rats that received fructose water for eight weeks had a significantly higher serum leptin level (2.837 ± 0.17 ng/mL) compared to the control group (1.072 ± 0.09 ng/mL) (*p* < 0.05). The treatment period of 28 days showed that FDW-PS and FDW-NGN rats had a significantly lower level of leptin (1.653 ± 0.06 ng/mL and 2.070 ± 0.17 ng/mL, respectively) compared to FDW rats (*p* < 0.05). In addition, FDW-PS rats had a more significant reduction in leptin level compared to FDW-NGN rats (*p* < 0.05). Adiponectin levels in FDW rats (15.36 ± 1.18 ng/mL) were significantly lower because of the fructose induction compared to the control group (45.63 ± 3.19 ng/mL) (*p* < 0.05). However, FDW-PS and FDW-NGN rats showed a significant increment in the levels of adiponectin (33.35 ± 0.62 ng/mL and 25.39 ± 0.51 ng/mL, respectively) compared to the FDW group (*p* < 0.05). Furthermore, the increment of adiponectin level was more significant in FDW-PS rats compared to FDW-NGN rats (*p* < 0.05) (Figure 3).

#### 2.2.4. HMG-CoA Reductase Activity

Induction of fructose for eight weeks significantly increased the HMG-CoA reductase activity in FDW rats (4.470 ± 0.19 ng/mL) compared to the control group (0.987 ± 0.04 ng/mL), whereas the FDW-PS and FDW-NGN group rats significantly lowered the increase in HMG-CoA activity (3.187 ± 0.10 ng/mL), (3.864 ± 0.12 ng/mL) compared to the FDW group. However, FDW-PS showed a more significant reduction in the HMG-CoA activity compared to the FDW-NGN group rats (Figure 4).

#### 2.2.5. Histological Analysis

Processed adipose tissue was stained with H&E staining for microscopic study. No differences in adipose tissue morphology were observed between the baseline group and control group rats (Figure 5A,B). Adipocyte hypertrophy was seen in the FDW group rats with higher adipocyte surface area (µm^2^) compared to the control group (Figure 5B,C). Treatment with PS and naringin reduced the adipocyte hypertrophy compared to the FDW group (Figure 5D,E).

#### 2.2.6. Lipid Profiles

There were no significant differences between the control group and baseline group in their lipid levels. The FDW group showed a significantly higher serum total cholesterol, triglyceride, LDL with significantly lower HDL levels compared to the control group (*p* < 0.05), following the induction of fructose. After 28 days of treatment, FDW-PS and FDW-NGN rats showed a significant reduction in serum total cholesterol, triglycerides, and LDL levels with significantly higher levels of serum HDL compared to the FDW group (*p* < 0.05). The reduction in total cholesterol, triglycerides, and LDL levels were lower in the FDW-PS rats compared to the FDW-NGN rats (*p* < 0.05). However, there were no significant differences in the increment of HDL levels between the FDW-PS and FDW-NGN groups (Figure 6).

## 3. Discussion

Lifestyle modifications that include proper diet and adequate physical activity is the first-line treatment for metabolic syndrome. Over the years, studies have shown promising benefits of medicinal plants toward metabolic syndrome. These favorable benefits were thought to be related to their bioactive substances that reduce the risk of developing cardiovascular diseases. Hence, the medicinal benefits of natural plants have become the focus in a lot of ethnopharmacology research. This study investigated the anti-obesity and lipid-lowering effects of *Piper sarmentosum* methanolic extract in fructose-induced obese rats for a period of 12 weeks. The consumption of fructose water displayed a tremendous increase in body weight, fat mass, and percentage fat. These indicators of obesity exhibited the development and progression of obesity in the rats fed with fructose. As fructose bypasses the rate-limiting steps of glucose metabolism, fructose metabolism is rapid and continuous, leading to excessive fatty acid synthesis that is further assembled to form triglycerides [18].

The excessive amount of fatty acids and triglycerides are then stored in the lipid droplets of fat cells around the body. This leads to the generation of excess fat in the visceral and subcutaneous regions, which accounts for the increase in body weight, fat mass, and percentage fat as shown in this study. However, treating obese rats with methanolic extract of *Piper sarmentosum* reduced the increment of body weight, fat mass, and percentage fat. In addition, these results are comparable to the positive control of the naringin treated group. However, treatment with *Piper sarmentosum* had a more eminent effect compared to naringin. The possible mechanism involved in the reduction in body weight, fat mass, and percentage age fat is through the inhibition of pancreatic lipase enzyme. Pancreatic lipase is an enzyme that catalyzes the hydrolysis of triacylglycerol into mono- and diacylglycerol and fatty acids [19]. Excessive generation of these end products would be absorbed into the body, which is responsible for the development of obesity [20]. Studies have shown that plant compounds are able to suppress the activity of this enzyme, hence would reduce the rate conversion of triacylglycerol. Therefore, compounds such as polyphenols and flavonoids present in the methanolic extract of *Piper sarmentosum* [21] may cause a synergetic inhibitory activity against pancreatic lipase. This then led to a decrease in the absorption of fats in the form of triglycerides, resulting in weight loss [22].

Adipocytes are fat cells that are primarily composed of adipose tissue. Adipocytes are major energy storage sites and have crucial endocrine functions. Excessive adipose tissue accumulation causes hormonal and metabolic changes that contribute to obesity and metabolic diseases. The fundamental reason behind the accumulation of adipose tissue is due to the imbalance between calorie intake and energy expenditure in which the excess calorie is stored as fat in the body. In this study, the histological findings revealed the phenomenon of adipocyte hypertrophy in the rats supplemented with fructose for 12 weeks. This could be that the surface area (µm^2^) of adipocytes became enlarged compared to the control group. The mechanism that underlies this phenomenon is that fructose consumption increased the adipogenic potential on the adipocyte precursor cells, hence accelerating adipocyte hypertrophy [23].

Treatment of the obese rats with an methanolic extract of *Piper sarmentosum* at a dosage of 125 mg/kg led to hypotrophy of adipocytes in which the adipocyte cells shrunk in size. In addition, these results are comparable to the naringin treated group as it also led to adipocyte shrinkage. There are several possible mechanisms involved in the suppression of adipocytes by *Piper sarmentosum*, which includes the inhibition of pancreatic lipases and enhancement of lipid metabolism (lipolysis) through induction of β-oxidation secretion in fats cells, promoting energy expenditure by increasing the metabolic rate and by decreasing the appetite [24]. These mechanisms would cause a lower amount of fats to be stored in the adipocytes and a greater amount of fats to be eliminated from the body through oily feces. This further indicates that the methanolic extract of *Piper sarmentosum* inhibited adipogenesis and the formation of fat cells in the adipose tissues, further reducing the size of the adipocyte.

Adipokines are signaling molecules secreted by adipose tissue to communicate with other organs including the brain, liver, muscle, immune system, and adipose tissue itself [25]. The commonly known adipokines are leptin and adiponectin. Leptin function is to inhibit appetite, stimulate thermogenesis, enhance fatty acid oxidation, decrease glucose, and reduce body weight and fat [26]. Adiponectin is an insulin sensitizing hormone that plays an important role in the modulation of glucose and lipid metabolism. Excess nutrition and a sedentary lifestyle induce excess fat to be stored in the adipose tissue and peripheral tissues [26]. This excess fat storage initiates adipocyte hypertrophy and hyperplasia [25]. The enlargement of adipocytes provokes dysregulation in the expression and secretion of adipokines [25]. In this study, fructose consumption by rats significantly elevated the levels of leptin, reducing the adiponectin levels. The mechanism involved in the increment of leptin levels is attributed to the excessive secretion of leptin that contributes to leptin resistance. These elevated levels of leptin fail to control hunger and modulate body weight. Apart from the elevated leptin level, adiponectin level was also diminished because of fructose consumption. This may suggest that fructose consumption altered the expression of adiponectin, leading to lower secretion of adipokines.

Treating obese rats with a methanolic extract of *Piper sarmentosum* at a dosage of 125 mg/kg led to the reduction in the elevated level of leptin compared to the control group. This result is comparable to the naringin treated group, which elicited similar effects toward leptin levels. However, the reduction in leptin levels by *Piper sarmentosum* treatment had a more potent effect compared to the naringin treated group. The mechanism underlying the reduction in the leptin level might be due to the anti-obesity effect of *Piper sarmentosum*. Circulating leptin levels is proportional to the general adiposity and reduced adipose fat can result in decreased leptin concentrations [27]. As the methanolic extract of *Piper sarmentosum* suppressed and reduced the adipocytes in obese rats, this in return reduced the elevated levels of leptin. In addition, the extracts may have reduced the levels of leptin by restoring and improving the target cells’ leptin sensitivity through the modulation of the hypothalamic leptin signaling pathway [28]. Apart from that, treatment of the obese rats with a methanolic extract of *Piper sarmentosum* at a dosage of 125 mg/kg led to an increase in the diminished level of adiponectin. The extract of *Piper sarmentosum* enhanced the adiponectin gene expression that led to an increase in the adiponectin level. This enhanced level of adiponectin would stimulate insulin secretion, which would mediate weight loss.

HMG-CoA reductase is the rate-controlling enzyme of the mevalonate pathway and is responsible for the synthesis of cholesterol in the liver. The findings in this study revealed that fructose consumption for 12 weeks significantly increased the activity of HMG-CoA reductase, which in turn catalyzes the excessive production of cholesterol. These findings are similar to the research conducted by [29,30,31] in which fructose intake increased the activity and expression of HMG-CoA reductase in C57BL/6 mice and Wistar rats. Furthermore, [28] also added that the addition of fructose to cultured rat hepatocytes increased HMG-CoA reductase activity by 3-fold. Thus, the hyperactivity of HMG-CoA reductase leads to an excessive amount of endogenous cholesterol synthesis [29]. This study indicated that the activity of HMG-CoA reductase was significantly inhibited by the methanolic extract of *Piper sarmentosum*. This inhibitory effect of *Piper sarmentosum* is more effective compared to the naringin treated group rats. The inhibitory effect toward HMG-CoA reductase is due to the plant’s phytochemicals, which exerted statin-like effects that reduces cholesterol biosynthesis [32,33]. In order to strengthen the findings of this study, we further investigated the lipid-lowering effects of *Piper sarmentosum* in fructose-induced obese rats for 12 weeks. Peculiarly, the fructose-induced rats had an elevated level of LDL, triglycerides, total cholesterol, and lower levels of HDL. As discussed previously, the metabolism of fructose in the liver generates an excessive amount of fatty acid that is converted into triglycerides. These excess triglycerides are then transported into the blood by VLDL and to the tissues in the body for storage. As fructose metabolism is continuous, consumption of excessive fructose would thereby increase the level of circulating levels of LDL, triglycerides, and total cholesterol in the blood.

Treatment of the obese rats with a methanolic extract of *Piper sarmentosum* at a dosage of 125 mg/kg and the positive control naringin at a dosage of 100 mg/kg significantly reduced the enhanced level of LDL, triglycerides, and TC and significantly increased the level of HDL. However, the reduction in lipid levels elicited by *Piper sarmentosum* was more efficient than the positive control naringin. The decrease in the levels of LDL, triglycerides, and total cholesterol occurred due to the inhibition of HMG-CoA reductase in the liver. This led to the suppression of endogenous cholesterol synthesis, which then led to lower lipid levels in the blood. Furthermore, sterols present in *Piper sarmentosum* are responsible for the cholesterol-lowering effect, which then elicits the reduction in LDL, triglycerides, and TC and increment in the levels of HDL [34,35]. Apart from that, a different mechanism such as competition with cholesterol for solubilization in dietary mixed micelles, co-crystallization with cholesterol to form insoluble mixed crystals, and interference with the hydrolysis process by lipases and cholesterol esterases are believed to contribute to the lowering of serum cholesterol concentrations [36]. As a consequence of all these mechanisms of sterols, intestinal cholesterol absorption is reduced while more cholesterol is excreted in the feces.

The outcomes of this study in experimental Wistar rats could suggest that *Piper sarmentosum* may be effective as an anti-obesity and -hyperlipidemic agent although not directly applicable to humans. To the best of our knowledge, this is the first study that discusses the use of the methanolic extract of *Piper sarmentosum* on leptin levels and adipocyte suppression. Furthermore, this study also shows the effect of *Piper sarmentosum* toward the fructose-induced metabolic syndrome rat model as most previous studies have adopted a high-fat diet technique. A DXA scan was used to determine the mass and percentage of visceral fat that are able to provide more accurate data compared to a hand-operated method. Future studies need to determine the active compound that is responsible for its effectiveness toward obesity and hyperlipidemia. In addition, the treatment period with a methanolic extract of *Piper sarmentosum* should be increased in order to obtain a more eminent result.

## 4. Materials and Methods

### 4.1. Preparation of Piper Sarmentosum Roxb. [Piperaceae] (PS) Methanol Extract

Fresh leaves (1 kg) of *Piper sarmentosum* Roxb. [Piperaceae] (PS) were collected from the Ethnobotanic Garden, Forest Research Institute Malaysia (FRIM) after being identified and confirmed by a plant taxonomist from the Universiti Kebangsaan Malaysia’s Herbarium (voucher specimen, UKMB40433). The plant extraction procedures were performed at the FRIM laboratory. Briefly, fresh leaves of the plant were cleaned with tap water and dried at room temperature before being chopped and ground into powder form. The powdered plant was then suspended in 80% methanol and heated using a Soxhlet extractor at 40–60 °C. The resulting heated solution was filtered using Whatman filter paper to obtain the supernatant. The supernatant was stored for the evaporation process using the rotary evaporator to remove all the remaining methanol content. The remaining supernatant in liquid form was transferred to a dark reagent bottle and kept in the freezer at −20 °C until further use. The liquid chromatography mass spectrometry (LCMS) analysis of the plant extraction was done at the Research and Instrumentation Management Center (CRIM) UKM and the data on the active compounds and purity were published recently [37].

### 4.2. Animal Preparation

All procedures were carried out following the institutional guidelines for animal research of Universiti Kebangsaan Malaysia (FAR/PP/2017/NUR AZLINA/29-MARCH/835-APR.-2017-APR.-2019). Forty male Wistar rats weighing 250–300 grams were maintained in a temperature-controlled room (24 °C) with adequate ventilation and illuminated for 12 hours daily (lights on from 0700 to 1900). The rats were kept in a single cage and were allowed free access to food and water. The rats were randomly divided into five experimental groups. A baseline group was used to verify the homogeneity of the control and treatment groups’ rat at the beginning of the experiment. The control group was used to differentiate the variation during the fructose induction period after 12 weeks with the FDW group. Three groups underwent an experimental induction of fructose to develop metabolic syndrome. Rats that exhibited any three or more of the metabolic syndrome criteria were further used for the treatment period. Olive oil was used as a vehicle in this experiment, thus, no differences in valley intake. The FDW group received 0.1 mL/100 mg olive oil. The FDW-PS group received 125 mg/kg *Piper sarmentosum* extract diluted in 0.1/100 mg olive oil. FDW-NGN group received 100 mg/kg naringin extract diluted in 0.1/100 mg olive oil. The naringin treated group was added to compare the effect of the *Piper sarmentosum* extract with naringin on all parameters studied. The groups are as follows; BN: Baseline; CTRL: Control; FDW: Olive oil + fructose; FDW-PS: PS + fructose; FDW-NGN: Naringin + fructose. Each group consisted of an equal number of rats (*n* = 8).

### 4.3. Administration of Fructose Drinking Water

The fructose used was D-fructose > 99% supplied by Syarikat System Malaysia. In order to prepare the 20% fructose drinking water, 20 g of fructose was diluted in 100 mL of tap water. The prepared solution was transferred to the 200 mL drinking bottles with ball bearings. Bottles with ball bearings were used to prevent the fructose solution from dripping. The bottles were then covered with aluminum foil to prevent fermentation. The fructose drinking water was administered for 12 weeks ad libitum. The physiological and metabolic parameters of rats were measured accordingly.

### 4.4. Dosage of Treatment

The dosage of 125 mg/kg of PS was chosen to treat metabolic syndrome rats based on previous studies by Azlina et al. [38] and Ali et al. [32], where they reported that at the dose chosen, there was a significant reduction in LDL, TC, and triglycerides as well as a significant increment in the levels of adiponectin and HDL. In addition, Ali et al. [32] further reported that 125 mg/kg of PS could significantly reduce the activity of HMG-CoA reductase.

The dosage of 100 mg/kg naringin was chosen based on previous studies [39,40,41]. These studies reported that naringin significantly reduced non-esterified fatty acids (NEFA), triglycerides, LDL, TC, abdominal fat deposition, abdominal circumference, fat droplets and significantly reduced HDL levels.

### 4.5. Physiological Measurements

Daily food intake, fluid intake, and calorie intake were measured every day for 12 weeks and the mean was compared. The food and fluid intake were measured by subtracting the initial amount provided to the remaining amounts in the cage. The calorie intake was calculated based on the consumption of fluid and food intake.

### 4.6. Metabolic Parameters

Body weight, fat mass, percentage mass, adiponectin and leptin levels, histological analysis of adipose tissue as well as HMG-CoA reductase enzyme activity were measured as indicators of obesity. The blood biochemistry test (lipid profile test) was done as indicators of hyperlipidemia.

#### 4.6.1. Body Weight

Body weight was measured weekly for 12 weeks using an electronic weighing scale (Comanche CMKS-690-6kg).

#### 4.6.2. Fat Mass and Percentage

Fat mass and percentage were measured at day 0, week 8, and the end of the 12th week using a DXA scan. Before initiating the body scan, the rats were anesthetized with 0.1 mL/100 g Ketamil and 0.01 mL/100 g Xylaxin®. The whole body DXA scan for each rat required 240 s. The scan results were analyzed using the recommended small animal analysis software.

#### 4.6.3. Adipocytokines

Serum adiponectin (Rat ADP/Acrp30/ER0006) and leptin (Rat Lep/ER0115) levels were determined using the rat adiponectin and leptin enzyme-linked immunoabsorbent assay (ELISA) kit (Fine Test, Wuhan Fine Biological, Wuhan, China). The analysis was done using a microplate reader at 450 nm (Parkin Elmer, Waltham, MA, USA).

#### 4.6.4. HMG-CoA Reductase Activity

The hepatic enzyme activity of HMG-CoA reductase was measured as an indicator of cholesterol synthesis that leads to hyperlipidemia. HMG-CoA reductase activity was measured using the commercially available kit from Fine Test. The analysis was done using a microplate reader at 450 nm (Parkin Elmer, Waltham, MA, USA).

#### 4.6.5. Histological Analysis

The histological analysis was done after the sacrificial period (week 12). After the rats were sacrificed with anesthesia (Ketamil and Xylazin), a longitudinal incision was performed at the anterior region of the rat. The deposition of adipose tissue around the region was observed in situ and then removed. The retrieved adipose tissue was then immediately fixed in 10% formalin for three days. The tissues were processed accordingly and thin sections of about 5 µm were obtained to be placed on microscope slides. The tissue sections were then stained with hematoxylin and eosin (H&E) for histomorphometry of the adipocytes. Once the staining was done, the section was mounted on dibutyl phthalate in xylene (DPX). Histomorphometry of adipocytes was analyzed using a light microscope (Nikon Eclipse, Tokyo, Japan). The area size of the adipocytes was measured in cm^2^.

#### 4.6.6. Lipid Profiles

A lipid profile test was performed as an indicator of hyperlipidemia. The blood samples were attained at day 0, week 8, and the end of week 12 via an orbital vein in the anesthetized rats. The rats were fasted overnight with tap water before withdrawing the blood. The serum samples were sent to Prima Nexus for analysis of the lipid profile. The analysis was done using commercial kits from Siemens Healthcare Diagnostics.

### 4.7. Statistical Analysis

All statistical analysis was conducted with the Statistical Package for Social Sciences (SPSS, version 23) software. The data were tested for normal distribution and presented as mean ± standard deviation (SD). Statistical significance (*p* < 0.05) was analyzed by one-way ANOVA followed by the Tukey test for multiple group comparison.

## 5. Conclusions

Oral administration of 125 mg/kg methanolic extract of *Piper sarmentosum* to fructose-induced obese rats led to a significant amelioration of obesity and hyperlipidemia. These are due to the phytochemicals of the plants that exerted protective effects by suppressing adipocytes that reduce obesity and inhibit HMG-CoA, which reduces hyperlipidemia. In conclusion, *Piper sarmentosum* may be considered as an alternative therapy for obesity and hyperlipidemia.

## Figures and Tables

**Figure 1 molecules-26-03985-f001:**
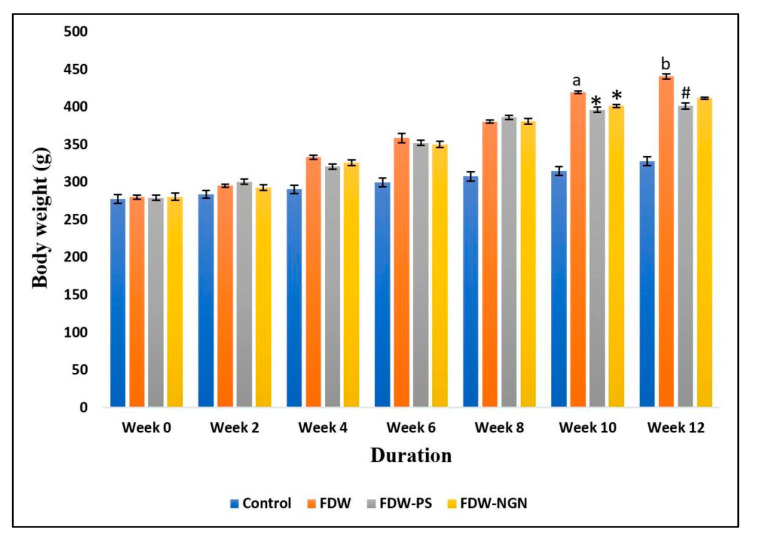
Body weight differences of all groups over 12 weeks. Values are mean ± SD and *n* = 8 for each group. Superscript letters indicate significantly different. ^a^ *p* < 0.05 compared to the control group at week 10. ^b^ *p* < 0.05 compared to the control group at week 12. * *p* < 0.05 compared to FDW at week 10. ^#^ *p* < 0.05 compared to FDW and FDW-NGN at week 12. FDW: Olive oil + 20% fructose; FDW-PS: PS + 20% fructose; FDW-NGN: Naringin + 20% fructose.

**Figure 2 molecules-26-03985-f002:**
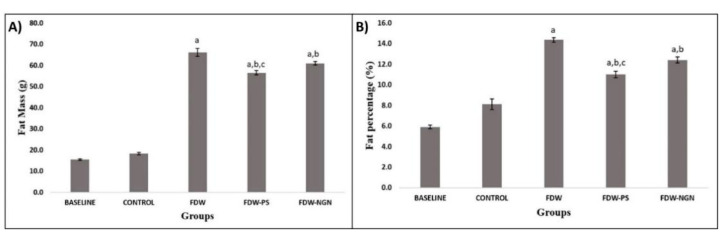
Fat mass (**A**) and fat percentage (**B**) differences of all groups over 12 weeks. Values are mean ± SD and *n* = 8 for each group. Superscript letters indicate significantly different. ^a^ *p* < 0.05 compared to the control group. ^b^ *p* < 0.05 compared to the FDW group. ^c^ *p* < 0.05 compared to the FDW-NGN group. FDW: Olive oil + 20% fructose; FDW-PS: PS + 20% fructose; FDW-NGN: Naringin + 20% fructose.

**Figure 3 molecules-26-03985-f003:**
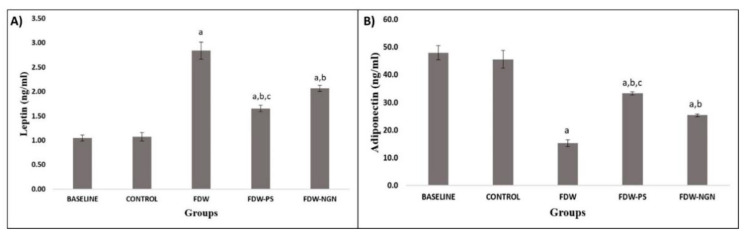
Leptin (**A**) and adiponectin (**B**) levels of all groups over 12 weeks. Values are mean ± SD and *n* = 8 for each group. Superscript letters indicate significantly different. ^a^ *p* < 0.05 compared to the control group. ^b^ *p* < 0.05 compared to the FDW group. ^c^ *p* < 0.05 compared to the FDW-NGN group. FDW: Olive oil + 20% fructose; FDW-PS: PS + 20% fructose; FDW-NGN: Naringin + 20% fructose.

**Figure 4 molecules-26-03985-f004:**
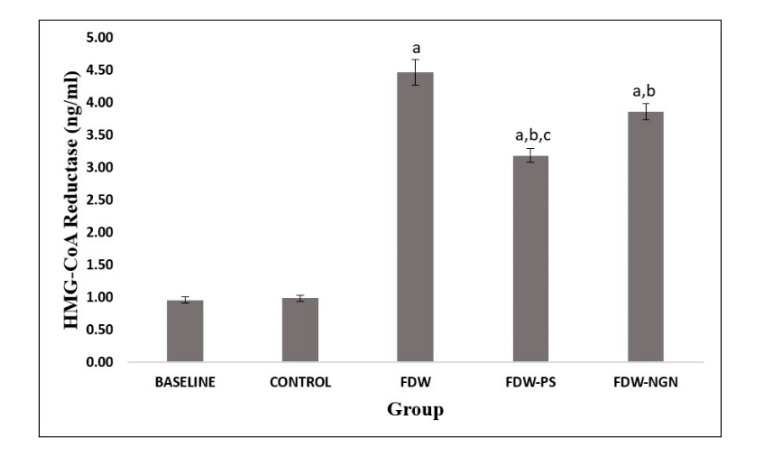
HMG-CoA reductase levels of all groups over 12 weeks. Values are mean ± SD and *n* = 8 for each group. Superscript letters indicate significantly different. ^a^ *p* < 0.05 compared to the control group. ^b^ *p* < 0.05 compared to the FDW group. ^c^ *p* < 0.05 compared to the FDW-NGN group. FDW: Olive oil + 20% fructose; FDW-PS: PS + 20% fructose; FDW-NGN: Naringin + 20% fructose.

**Figure 5 molecules-26-03985-f005:**
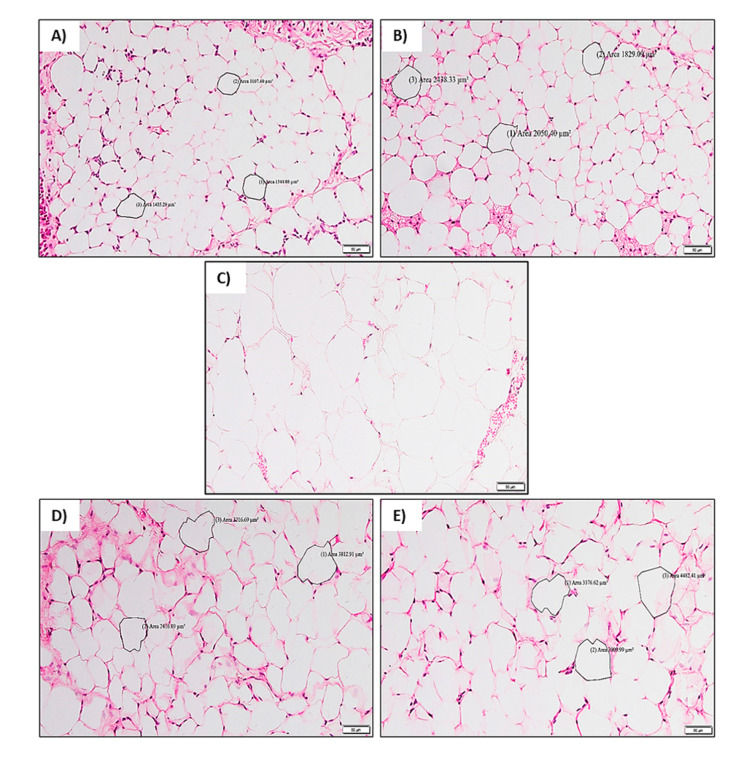
The histomorphometry of the adipocyte tissue of all groups. The size of the adipocytes was increased in the FDW group (**C**) compared to the baseline (**A**) and control (**B**) groups. The size of adipocytes of FDW-PS (**D**) and FDW-NGN (**E**) were reduced compared to FDW. The size of the adipocytes was measured and indicated using surface area (µm^2^). Scale bar represents 60 μm. FDW: Olive oil + 20% fructose; FDW-PS: PS + 20% fructose; FDW-NGN: Naringin + 20% fructose.

**Figure 6 molecules-26-03985-f006:**
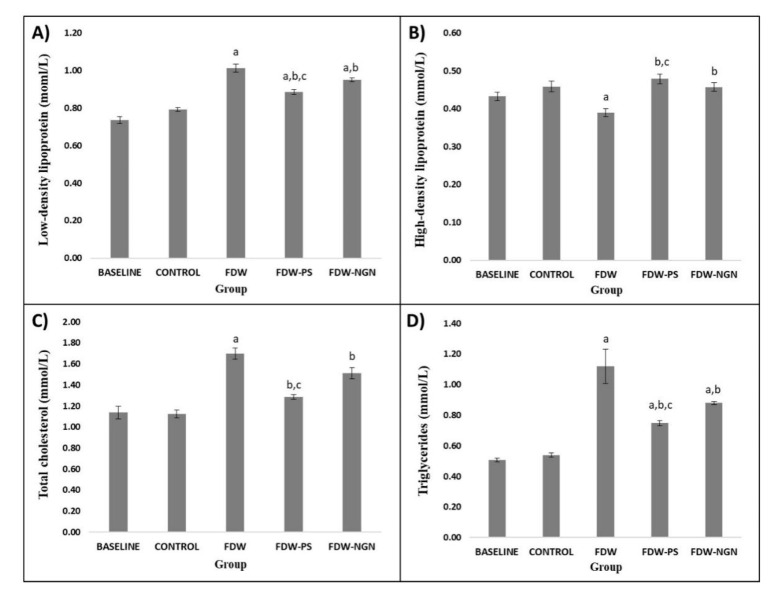
Lipid profiles comprising of low density lipoprotein (**A**), high density lipoprotein (**B**), total cholesterol (**C**), and triglycerides (**D**) of all groups over 12 weeks. Values are mean ± SD and *n* = 8 for each group. Superscript letters indicate significantly different. ^a^ *p* < 0.05 compared to the control group. ^b^ *p* < 0.05 compared to the FDW group. ^c^ *p* < 0.05 compared to the FDW-NGN group. FDW: Olive oil + 20% fructose; FDW-PS: PS + 20% fructose; FDW-NGN: Naringin + 20% fructose.

**Table 1 molecules-26-03985-t001:** Effects of fructose, *Piper sarmentosum*, and naringin on physiological variables of all groups over 12 weeks.

Variables	CTRL	FDW	FDW-PS	FDW-NGN
Food intake (g/day)	15.23 ± 0.54	20.91 ± 0.67 ^a^	18.15 ± 0.51 ^a,b^	17.55 ± 0.53 ^a,b^
Fluid intake (ml/day)	44.87 ± 1.12	51.75 ± 1.03 ^a^	49.75 ± 1.03 ^a,b^	50.5 ± 1.18 ^a,b^
Calorie intake (Kcal/day)	42.66 ± 1.53	100.3 ± 1.94 ^a,b^	90.6 ± 1.19 ^a,b^	88.9 ± 1.92 ^a,b^

Values are mean ± SD and *n* = 8 for each group. Superscript letters indicate significantly different. ^a^ *p* < 0.05 compared to CTRL group. ^b^ *p* < 0.05 compared to the FDW group. CTRL: Control; FDW: Olive oil + 20% fructose; FDW-PS: PS + 20% fructose; FDW-NGN: Naringin + 20% fructose.

## Data Availability

Data are contained within the article.

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
