# Peer review of "Methanolic Extract of Piper sarmentosum Attenuates Obesity and Hyperlipidemia in Fructose-Induced Metabolic Syndrome Rats"

_molecules, 2021, doi:10.3390/molecules26133985_

Round 1
Reviewer 1 Report
This study was attempted to investigate the effect of methanolic extract of Piper sarmentosum towards obesity and hy-perlipidemia in fructose-induced metabolic syndrome in rat model. However, some points have to be addressed.
- In the introduction section, a hypothesis should be added.
- Materials and methods
(1) L361: Add more details about the Baseline and Control group. Why olive oil was added in the FDW group? The information of Naringin should be added.
(2) Is there any standard for the metabolic syndrome model?
(3) L428: “p < 0.05 indicated significant differences” should be added.
- Results
- In figure 2-6, there was baseline group, but it was not appeared in figure 1 and table 1. Why?
- How about the visceral fat mass of all groups over 12 weeks?
- In the discussion section, strengths and weaknesses in relation to other studies should be included.
Reviewer 2 Report
The author shows an interesting manuscript about the effect of methanolic extract of PS as a bioactive vegetal product with beneficial properties reducing fat deposition, lipid levels, and fasting glucose, among others. The authors used an obesity fructose-induced model, and they found that PS improves lipid profile and adipocytokines levels. However, the central problem of the article is the experiment design.
Major concerns
- What is the difference between the baseline and control group? Please clarify.
- Concerning adipocytokines measurement, please provide more information about the kits used; ¿are they specific for rats?
- Authors must explain the election of the biomarkers used, why don´t author choose insulin or oral glucose tolerance, being the aim of this study demonstrate the effects of PS on metabolic syndrome (line 94). In the same line, measurement of HMG-CoA reductase activity is not a marker directly related to a metabolic syndrome induced by fructose. Dyslipidemia metabolic syndrome-associated is characterized by the increase of triglycerides instead of cholesterol. If authors have according referenced that fructose increase the HMG-CoA reductase activity, please provide them.
- Fig. 5. The histomorphometry must be joined with a quantitative analysis of the adipocyte size.
- Authors must be clarifying the groups' nomenclature. I understand that FDW is olive oil + 20% fructose; indeed, it is unnecessary to repeat 20% fructose for the rest of the groups; this is confusing. If only FDW was treated with olive oil, the groups are not comparable for many reasons (calory intake, antioxidants, sugar/lipid mix), and metabolic alterations in FDW probably were higher for this.
On the other part, why author use the term BN? Baseline is cero time in kinetics. The correct term for this group is untreated or control group.
- The paragraph in discussion: “The probable mechanism involved in the reduction in body weight, fat mass and percentage is through the inhibition of pancreatic lipase enzyme activity which is responsible for digestion and absorption of triglycerides” why the author says that? please use references to explain it.
- The phrase “The hepatic enzyme activity of HMG-CoA reductase was measured as an indicator of obesity” I don´t think is an obesity indicator; please clarify.
Minor
Line from 98 to 100 must be deleted.
Round 2
Reviewer 2 Report
I think that the manuscript rather improved about the primary concern. I don´t see anything else to comment on it.
Author Response
Dear Reviewer,
Thank you for your comments.
Regards,
Nurul Alimah Abdul Nasir